# Stress-Induced Membraneless Organelles in Neurons: Bridging Liquid–Liquid Phase Separation and Neurodevelopmental Dysfunction

**DOI:** 10.3390/ijms26189068

**Published:** 2025-09-17

**Authors:** Norbert Bencsik, Daniel Kimsanaliev, Krisztián Tárnok, Katalin Schlett

**Affiliations:** Department of Physiology and Neurobiology, Institute of Biology, Eötvös Loránd University, 1117 Budapest, Hungary; daniel.kimsanaliev@ttk.elte.hu (D.K.); tarnok.krisztian@ttk.elte.hu (K.T.)

**Keywords:** liquid–liquid phase separation (LLPS), neurodevelopmental disorder (NDD), stress granule, nuclear paraspeckle, P-body, intrinsically disordered region

## Abstract

Liquid–liquid phase separation (LLPS) in cell biology has revolutionized our understanding of how cells organize biochemical reactions and structures through dynamic, membraneless organelles (MLOs). In neurons, LLPS-driven processes are particularly important for regulating synaptic plasticity, RNA metabolism, and responses to environmental stressors. Over the past decade, LLPS has gained increasing attention in neurobiology as a framework to interpret altered synaptic functions in various neurodevelopmental disorders (NDDs). These diseases comprise a diverse spectrum of clinical and pathological symptoms (e.g., global developmental delay, impaired cognitive and mental functions, as well as social withdrawal). Recent studies have highlighted how mutations in proteins containing intrinsically disordered regions (IDRs)—key drivers of LLPS—can alter condensate properties, resulting in persistent or defective MLO formation. These aberrant assemblies may disrupt RNA transport, splicing, and translation in developing neurons, thereby contributing to disorder pathology. IDRs are known to be enriched in membraneless components, such as stress granules, nuclear paraspeckles, and P-bodies, where they play crucial role in the formation, maintenance, and function of protein–RNA networks. This review explores the role of stress-induced MLOs in the nervous system, the molecular principles governing their formation, and how their dysfunction bridges the gap between environmental stress responses and neurodevelopmental impairment.

## 1. Introduction

Neurodevelopmental disorders (NDDs) are a group of conditions that originate during the development of the central nervous system, typically emerging in infancy or early childhood during a sensitive period marked by rapid brain growth and high plasticity, when the brain is especially responsive to both positive and negative environmental influences [1,2]. Although the onset of NDDs is driven by diverse factors, including genetic inheritance and/or a wide range of environmental factors, NDDs share common features involving altered neural network development and function. From a genetic perspective, many NDDs have a hereditary component or are caused by spontaneous de novo mutations that impact neuronal development [3,4]. Environmental risk factors are equally critical and may include prenatal exposures to harmful substances, e.g., alcohol, tobacco, and medications, as well as to toxic chemicals (reviewed in [5]). Maternal infections during pregnancy (e.g., Rubella and Zika viruses) can also interfere with fetal development and increase the risk of conditions such as microcephaly or developmental delays [6,7]. Additionally, perinatal complications, for instance, premature birth, low birth weight, and oxygen deprivation, are also known to elevate the risk of developing NDDs [8].

NDDs can vary in severity, ranging from mild learning disabilities to profound intellectual and behavioral challenges (e.g., global developmental delay, impaired cognitive and mental functions, as well as social avoidance and withdrawal). Corresponding communication disorders, intellectual disabilities, autism spectrum disorders (ASD), attention deficit hyperactivity disorders (ADHD), motor disorders, or specific learning disorders are responsible for serious health and mental issues worldwide [9] and pose a substantial burden to society. The global prevalence of NDDs varies widely due to the largely varying methodology of the studies, such as the different types of evaluation and characterization of affected children, as well as the level of activity of the countries in the detection and diagnosis of NDDs [9]. Neurodevelopmental disorders showed the highest estimated prevalence among children younger than 15 years, varying from 4.1% in those under 5 years old to 7.0% in the 10 to 14 years age group [10]. Differences in socioeconomic status, community types (rural or urban), ethnicity, and healthcare availability may influence prevalence, often resulting in underdiagnosis or overdiagnosis within certain populations. Interestingly, there is a strong comorbidity between fragile X syndrome and ASD; the prevalence of ASD in fragile X syndrome has been estimated at 50% [11,12]. Despite the intensive research and funding, the prevalent causes of NDDs are still mostly unknown. Malfunctions in molecular mechanisms, such as the Wnt/β-catenin [13], NOTCH3 [14], and SHH pathways [15], controlling neurogenesis and neuronal migration, resulting in neurons being mispositioned, are recognized as a key factor of various NDDs. Additionally, recent studies have highlighted a strong link between ASD and increased oxidative stress (reviewed in [16]).

In addition to these findings, several genes associated with NDDs have been found to encode proteins involved in liquid–liquid phase separation (LLPS), a fundamental cellular mechanism in neurons as well as in other cell types [17,18]. LLPS is primarily driven by weak and multivalent molecular interactions, including protein–protein, protein–RNA, and even RNA–RNA interactions. In this process, a uniform mixture of biomolecules separates into two distinct phases: one enriched in specific macromolecules and the other more diluted surrounding the dense region [19,20]. This mechanism facilitates the formation of membraneless organelles (MLOs), such as the nucleoli, Cajal body, super elongation complex (SEC) body, stress granules, nuclear paraspeckles, and P-bodies (Figure 1). MLOs are essential for compartmentalizing biochemical reactions without lipid bilayers. These structures enable the localized regulation of processes in the neurons, such as synaptic vesicles clustering [21,22], postsynaptic scaffolding [23,24], mRNA translation, RNA transport, and protein synthesis [25] at synapses. These functions are critical for synaptic plasticity, memory formation, and neuronal development [18]. Emerging scientific evidence suggests that abnormal regulation of MLO formation via LLPS is associated with various disorders [17], including NDDs, neurodegenerative diseases (e.g., amyotrophic lateral sclerosis, Alzheimer’s disease, and Parkinson’s disease) [26], and central nervous system tumors [27].

LLPS is driven by multivalent interactions among proteins or between proteins and RNA [28,29]. These interactions often involve weak, transient forces e.g., charge–charge interactions, π–π stacking, hydrogen bonding, and cation–π interactions. Specific molecular interactions among proteins and RNAs help to stabilize these structures, thus providing the resulting condensates’ form and function. Condensates can also incorporate regulatory elements driving their behavior and responsiveness.

Many proteins that undergo LLPS contain IDRs, which are sequence motifs lacking a stable three-dimensional structure. These areas often include low-complexity regions, which are rich in specific amino acids (e.g., glycine, proline, and alanine). For example, low-complexity regions can modulate LLPS behavior and serve protective roles by reducing proteotoxic stress [17]. Other examples of LLPS involve interactions between proline-rich motifs (PRMs) and their SRC Homology 3 (SH3) domains [30]. Both IDRs and SH3/PRM regulate engagement in multivalent interactions, which are essential for the formation and stability of phase-separated structures.

One of key advantages of LLPS is its dynamic and reversible nature, allowing condensates to adapt in response to environmental cues. Variations in factors such as salt concentrations, pH, protein abundance [31,32,33], or post-translational modifications (e.g., lysine acetylation, serine/threonine/tyrosine phosphorylation, and arginine methylation) can initiate the formation or dissolution of phase-separated structures [34,35]. For example, stress granules (SGs) form in response to cellular stress and dissolve once physiological conditions are restored, allowing neurons to resume protein synthesis and maintain mRNA stability [32].

Cellular stress is a potent trigger for the formation of the reversible MLOs, often referred to as stress-induced membraneless organelles. Due to their post-mitotic nature and high metabolic demands, neurons are particularly susceptible to stressors like oxidative damage, heat shock, and nutrient deprivation [36,37]. In response, neurons rapidly reorganize their intracellular environment by assembling dynamic MLOs. This adaptive mechanism enables neurons to prioritize the repair and stabilization of unfolded or misfolded proteins and RNAs and to prevent the accumulation of toxic intermediates. Importantly, the failure to properly assemble and/or disassemble stress-induced MLOs has been linked to persistent protein aggregation and to the onset of NDDs in neurons [18]. While many types of MLOs exist, stress-induced MLOs appear to play a particularly important role in NDDs.

In this review, we discuss the formation and regulation of the three most well-characterized stress assemblies: (i) stress granules [38], (ii) nuclear paraspeckles [39], and (iii) the RNA-based processing bodies (P-bodies) [40], which are formed in response to various insults such as heat, osmotic and oxidative stress, and numerous other factors [41]. We will review their physiological roles during neurodevelopment and the consequences of their dysregulation in the context of neurodevelopmental disorders. We also collect and highlight phase-separating proteins that contribute to the formation and pathology of neurodevelopmental disorders.

## 2. Stress Granules

Stress granules (SGs) are a distinct type of ribonucleoprotein granules, appearing as temporary cytosolic structures ranging from 0.1 to 2 μm in size (Figure 1). Importantly, SGs are non-membranous cytosolic organelles that form when translation initiation is inhibited. They are frequently associated with pathological granules in various NDDs [37,42,43]. Recently, RNA self-assembly was shown to contribute to the formation of phase-separated SGs. Studies of SG have revealed that they consist of two regionally distinct components: a stable “core” region, indicating that these assemblies are not purely formed via LLPS, and a less concentrated “shell” region, which behaves like an LLPS-mediated structure formed by weak interactions [33,44]. The mechanism underlying SG assembly remains poorly defined [45], although some insights have emerged from transcriptome analyses of SGs [46] and single-molecule imaging of RNA-binding proteins [47].

An initial aggregation step of stress granules can be triggered by the intermolecular interactions of ribonucleoproteins forming stable core structures. **G3BP1** (Ras-**G**AP SH**3** domain **b**inding **p**rotein**1**—G3BP1) is most well-studied for its role in SG initiation (Figure 1). Several studies have implicated G3BP1 as a central hub and key regulator of SG assembly. It plays a pivotal role in organizing core and shell regions: its interactions with RNA and other SG components mediate the shell’s rapid exchange dynamics, while multivalent interactions—particularly via IDRs and the RG-rich domain—contribute to the formation of dense cores [48,49]. Thus, the intrinsic properties of G3BP1 act as a nucleator that regulates protein–RNA interactions driving the formation LLPS (Table 1). Under stress conditions (e.g., axonal injuries), the phosphorylation of two serine residues, Ser-149 and Ser-232, within IDR1 promotes an open conformation of G3BP1. This conformational change exposes the RNA recognition motif (RRM) and the arginine/glycine-rich domain (RGG), thereby enhancing G3BP1-mediated, RNA-dependent LLPS [48,49,50].

Several stress granule-associated proteins, including G3BP1, have been linked to axonal outgrowth. The binding of G3BP1 to mRNA, shaping the structure of SGs, is thought to suppress the accumulation of free mRNA, which can support the neuron’s survival following axonal injury. Thus, axonal G3BP1 is likely a negative regulator of axonal protein synthesis and axon growth [51]. In addition, inactivation of the *G3BP1* gene results in embryonic lethality and neurodevelopmental abnormalities in some brain regions, including hippocampal CA1 pyramidal neurons, cortical neurons, and neurons of the internal capsule [52]. Martin and colleagues developed a viable G3BP1 knockout mouse model that exhibited behavioral defects associated with the central nervous system, including ataxia and ASD-like symptoms. Electrophysiological analyses revealed that the absence of G3BP1 enhances long-term depression and short-term potentiation in the hippocampal CA1 region. It is likely that the loss of G3BP1 protein in neurons leads to increased intracellular calcium release in response to metabotropic glutamate receptor activation [53]. Given these neurodevelopmental abnormalities, the link between axonal protein synthesis and calcium homeostasis is of particular interest.

Analysis of over 40,000 NDD cases revealed a significant enrichment of de novo variants in core SG-associated genes, including *G3BP1, G3BP2,* and *UBAP2L*. Many of these variants affect critical regions—such as the NTF2-like (nuclear transport factor 2—NTF2L) domain and IDRs—which are essential for SG assembly. Functional studies have confirmed that NDD-linked variants disrupt SG protein interactions and granule formation, supporting the idea of a shared SG-related pathology across multiple NDDs [54]. Multiple intrinsically disordered regions have also been identified in **UBAP2L** (**ub**iquitin-**a**ssociated **p**rotein **2**-**l**ike) that are likely to undergo LLPS (Table 1), along with several nuclear localization and export signals (Figure 1). UBAP2L may form liquid-like condensates through its disordered RGG/FG domains, thereby initiating SG core formation and coordinating interactions with P-bodies [55,56].

**TIA-1** (**T**-cell **i**ntracellular **a**ntigen-**1**) is a well-characterized stress granule component, and its recruitment into SGs is associated with phase separation (Figure 1) [57]. In vitro studies support the concept that zinc ions promote multimerization and phase separation of TIA-1, thereby driving the assembly of TIA-1-positive SGs [58]. Recent findings have shown that the low-complexity domain of TIA-1 in the absence of the RRM domain is unable to undergo phase separation in vitro, even in the presence of RNA. This region of TIA1 influences both its phase separation behavior and the physical properties of TIA1-driven LLPS assemblies, with these effects being dependent on proline residues (Table 1). Detailed biophysical analysis of proline-related variants have revealed that several disease-associated mutations enhance TIA1 aggregation and impair SG clearance [59].

TIA-1 is also a critical effector within a network of immune-related genes that have additional roles in synaptic plasticity and behavior. While TIA-1 KO pups display normal hippocampal cytoarchitecture in both female and male mice, the functional role of TIA-1 in hippocampus-dependent behavior appears to be restricted to the processing or storage of aversive, stress-dependent memories. These findings may be relevant to fear-related NDDs, such as post-traumatic stress disorder (PTSD) and anxiety disorders [60]. TIA-1 also plays an important role in regulating mRNA metabolism during the early stages of human neurodevelopment. Its regulatory influence is the strongest during the initial developmental phases and progressively declines as cells differentiate into neurons [61].

**CAPRIN1** (**c**ytoplasmic **a**ctivation/**pr**ol**i**feration—associated protei**n 1**) has also been shown to promote stress granule formation (Figure 1; Table 1). The intrinsically disordered regions of CAPRIN1 contain multiple sites for regulatory phosphorylation, which can modulate phase separation in vitro, thereby regulating the inhibition of mRNA translation [62,63]. A newly identified circular RNA associated with hepatocellular carcinoma, known as circVAMP3, originates from exon 3 and exon 4 of the *VAMP3* gene. Mechanistically, circVAMP3 appears to exert its tumor-suppressive effects by interacting with the CAPRIN1–G3BP1 complex, indicating that circVAMP3 may promote LLPS by scaffolding CAPRIN1 proteins [64]. CAPRIN1 is a component of the same ribosome-containing granules as synaptic functional regulator FMR1 (FMRP), implying a role in translational regulation similar to that of the FMRP family.

**Table 1 ijms-26-09068-t001:** Proteins undergoing phase separation linked to NDD.

Membraneless Organelle	UniProt Identifier	Protein Name	Protein Function	Type of NDD	Data Supporting NDD	Data Supporting LLPS	Protein Region(s) Mediating LLPS
stress granule	Q13283	G3BP1	DNA helicase	ataxia phenotype, ASD-like behavior	[52,53]	[48,49,50]	1–142 NTFL2 domain; 143–226 IDR1; 410–466 IDR3
stress granule	Q14157	UBAP2L	ubiquitin-specific protease	speech–language problems, intellectual disability	[54]	[55,56]	disordered RGG/FG domains
stress granule	P31483	TIA1	RNA-binding protein	PTSD, anxiety disorder	[60,61]	[58,59]	3 RRM domains and low-complexity regions
stress granule	Q14444	CAPRIN1	RNA-binding protein	fragile X-syndrome, autism spectrum disorder, ADHD, language delays	[65,66,67]	[62,64]	C-terminal low-complexity, disordered region of CAPRIN1
stress granule	Q06787	FMRP	RNA-binding protein	fragile X-syndrome, ASD	[68]	[69]	445–632 C-terminal R/G-rich RGG motif-containing LC region
stress granule	O00571	DDX3X	RNA helicase	intellectual disability, ASD-like phenotype; movement disorder	[70,71,72]	[34]	1–168 N-terminal S/K-rich low-complexity region IDR containing RG motifs and the eIF4E-binding motif
stress granule	P09651	hnRNPA1	RNA-binding protein	ASID/autism spectrum—intellectual disability	[73]	[32,74,75]	186–372 C-terminal G-rich prion-like low-complexity region
stress granule	Q01844	EWSR1	RNA-binding protein	ASD-like behavior	[76]	[77]	1–285 N-terminal S/Y/Q/G-rich disordered domain; 286–360 disordered RGG repeats; 361–447 RNA binding region RRM
nuclear paraspeckle	Q15233	NONO	RNA-binding protein	intellectual disability, global developmental delay	[78,79,80]	[81]	218–272 NOPS domain for homodimerization or heterodimerization with SFPQ
nuclear paraspeckle	Q96PK6	RBM14	RNA-binding protein	autism spectrum disorder	[82]	[83]	350–669 prion-like domain with 21 Y[G/N/A/S]AQ or [S/G]YG motifs
nuclear paraspeckle	P23246	SFPQ	RNA-binding protein	autism spectrum disorder	[84]	[81,85]	GPM-rich disordered region
P-body	Q9UPQ9	TNRC6B	RNA-binding protein	ASD, ADHD, developmental delay, intellectual disability	[86]	[87]	437–1056 disordered GW-rich N-terminal Argonaute binding domain with tryptophan residues in motifs I and II

CAPRIN1 localizes distinct granules in dendrites, and this appears to have specialized functions in neurons, e.g., in dendrite and dendritic spine development [68]. Deletion of CAPRIN1 leads to reduced sociality, impaired response to novelty, and decreased cognitive flexibility, traits associated with ASD-like behavior [65]. Furthermore, conditional deletion of CAPRIN1 in mice results in significant impairment of long-term memory formation [88]. Additionally, CAPRIN1 haploinsufficiency has been linked to an autosomal dominant NDD characterized by language delays, ADHD, intellectual or learning disabilities, and ASD. Additional associated symptoms may include abnormalities in the respiratory, skeletal, ocular, auditory, and gastrointestinal tract [66,67].

Additionally, CAPRIN 1 colocalizes in stress granules with the highly expressed RNA-binding protein **FMRP** (synaptic functional regulator **FMR**1 **p**rotein; see Figure 1 and Table 1). FMRP is a multifunctional protein involved in various cellular processes [68]. FMRP can regulate RNA synthesis by either controlling the expression of chromatin-modifying enzymes or the activity of transcription factors [89]. FMRP can also bind mRNAs and may modulate their translation, stability, or intracellular transport [90]. Last but not least, FMRP also directly interacts with proteins (e.g., ion channels) and regulates their activity [91]. Phosphorylation of the C-terminal IDR of FMRP promotes its phase separation, which correlates with in vitro translation inhibition [63,69]. Disturbance of these specific steps in SG dynamics can lead to the formation of aberrant granules and thereby contribute to NDDs. Importantly, the IDRs of FMRP and CAPRIN1 both contain several phosphorylation sites known to regulate LLPS in vitro, leading to control of de-adenylation and translation activities [62,63].

**DDX3X** (**D**EA**D** bo**x** RNA helicase **3**, **X** chromosome-linked) encodes an ATP-dependent RNA helicase of the DEAD box family (Figure 1; Table 1). DDX3X is implicated in mRNA metabolism, including neuronal transport [92], as well as the formation of cytoplasmic stress granules [93,94]. DEAD-box helicases act as key regulators of LLPS in neurons (reviewed in [71,95]). The spatiotemporal acetylation of the DDX3X intrinsically disordered region 1, modulated by the lysine acetyltransferase CBP (CREB-binding protein) and histone deacetylase 6 (HDAC6), plays a key role in regulating SG assembly. Upon stress, lysine acetyltransferase CBP undergoes auto-acetylation and becomes activated, triggering widespread protein acetylation, including that of DDX3X. Subsequently, HDAC6-mediated deacetylation of DDX3X is required to enable effective LLPS and proper SG formation [34].

DDX3X is frequently mutated in certain neurodevelopmental disorders. Detailed analyses of DDX3X missense mutations reveal impaired mRNA release, reduced helicase activity, altered translation of some targets, and inability to induce stress granule formation. Thus, it is not surprising that DDX3X is essential during cortical neurogenesis [96]. Patients with the rare DDX3X syndrome show a wide range of clinical symptoms of neurological and ophthalmological defects, including corpus callosum hypoplasia, polymicrogyria, and ventricular enlargement. Consistent with these findings, DDX3X is strongly implicated in behavioral abnormalities, developmental delay, intellectual disability, movement disorders, and ASD-like phenotypes [70,97]. The phenotypic spectrum of DDX3X syndrome is a recently identified genetic disorder primarily affecting females and only rarely observed in males [97,98]. Notably, Kennis et al. reported the first large-scale systematic analysis of affected males, providing detailed male-specific phenotypic correlations and family counseling [72].

During the secondary aggregation event, SG nucleators mediate diverse interactions among numerous SG-associated proteins, resulting in the formation of SGs and organization of the shell region (Figure 1). Proteins recruited to the shell region include **h**eterogeneous **n**uclear **r**ibo**n**ucleo**p**roteins (hnRNPs) and RNA-binding protein EWS (EWSR1—**Ew**ing **s**arcoma breakpoint **r**egion **1**) [37,42,43].

Although broadly expressed across different tissues, many hnRNPs display dynamic, developmentally stage-specific expression patterns in the brain, with particularly high levels during embryonic and early cortical development [99]. **hnRNPA1** is regarded to exert its key functions in cellular stress response and in neuronal dysfunction via regulating mRNA metabolism (Figure 1, Table 1). The C-terminal low-complexity region of hnRNPA1 mediates LLPS in vitro and is sufficient for recruitment into SGs in HeLa cells [32]. Specifically, the LLPS behavior of the hnRNPA1 C-terminal domain is modulated by electrostatic interactions influenced by changes in pH, as well as protein, salt, and RNA concentrations [32,74]. The interaction between hnRNPA1 and specific single-stranded RNA, as well as its involvement in LLPS, has also been investigated in vitro using magnetic resonance techniques. These studies revealed that the low-complexity domain adopts compact conformation and interacts with the RNA recognition motifs [75]. Missense mutations of hnRNPA genes have been identified in neurodegenerative diseases, including amyotrophic lateral sclerosis and multisystem proteinopathy [100]. In addition, individuals with hnRNPs-related syndrome show a high prevalence of intellectual disability, speech delay, and ASD and/or other neurodevelopmental phenotypes [73,101,102].

**EWSR1** (**Ew**ing **s**arcoma breakpoint **r**egion **1**) belongs to the FET family (**F**US, **E**WSR1, **T**AF15), a group of RNA-binding proteins involved in mRNA transport and alternative splicing (Figure 1, Table 1; [103]). A recent proteomic study of neural progenitors derived from three ASD affected individuals identified EWSR1 as one of the differentially expressed proteins in ASD neural progenitor cells [76]. Interestingly, EWSR1-knockout mice exhibit reduced neuronal nuclear size in the motor cortex, striatum, and hippocampus, accompanied by motor coordination abnormalities such as increased limb clasping and hyperkinesia [104]. Phase separation of EWSR1 is driven by cooperative interactions between tyrosine residues in its prion-like domain and arginine residues in the RNA-binding domain [77]. Notably, LLPS behavior of EWSR1 differs from that of the FUS protein, highlighting distinct functional properties within the FET protein family [105].

In summary, stress granules are dynamic, membraneless organelles formed via liquid–liquid phase separation, and they are crucial in regulating mRNA metabolism under cellular stress, particularly in neurons. Key proteins such as G3BP1, UBAP2L, TIA-1, CAPRIN1, FMRP, DDX3X, hnRNPA1, and EWSR1 coordinate SG assembly and function. Mutations in stress granule-associated genes often affect intrinsically disordered regions, disrupting LLPS capability and leading to neurodevelopmental disorder-related phenotypes, such as ASD, intellectual disability, and behavioral abnormalities.

## 3. Nuclear Paraspeckles

Nuclear paraspeckles are specialized structures composed of mRNAs and proteins that regulate gene expression. They share several structural and functional similarities with stress granules. Notably, paraspeckles are built around a long-noncoding RNA (lncRNA) molecule called NEAT1/2 (nuclear paraspeckle assembly transcript 1/2), which serves as an essential architectural scaffold for protein binding (Figure 1). Nuclear paraspeckles become more abundant during cellular stress, where they sequester various proteins and mRNAs, thereby modulating their functions [39,106,107]. Like other biomolecular condensates, the assembly of nuclear paraspeckles depends on a network of multivalent protein–protein and protein–RNA interactions. The formation of paraspeckles is driven by nuclear paraspeckle proteins, such as PSPC1. Although variations in PSPC1 have been associated with several diseases, they have not yet been directly linked to neurodevelopmental disorders as a primary diagnosis.

**NONO** (**non**-POU domain-containing **o**ctamer-binding protein) belongs to the highly conserved DBHS (Drosophila behavior/human splicing) protein family, which also includes SFPQ (**s**plicing **f**actor **p**roline/glutamine(**Q**)-rich) and PSPC1 (paraspeckle protein component 1) (Figure 1, Table 1). NONO and SFPQ are critical for nuclear paraspeckle formation, both by maintaining NEAT1/2 levels and by actively participating in paraspeckle assembling. Initially, NONO and SFPQ form a large heterodimer in the nucleus, mediated by the NOPS domain (NonA/paraspeckle domain) of NONO, which enables its dimerization with SFPQ via LLPS. Subsequently, the coiled-coil domain (CCD) extends outward from the globular dimer core of the PSPs, providing a platform for further polymerization. This action is driven by NONO/SFPQ interactions, promotes LLPS, and facilitates the formation of nuclear paraspeckles [81,108]. The later recruitment of the essential **RBM14** (**R**NA-**b**inding **m**otif protein **14**) further enhances LLPS, contributing to the complete assembly of the paraspeckle scaffold (Figure 1, Table 1) [83]. Meta-analyses have validated known and identified novel autism risk genes in a previously unstudied Brazilian cohort [82].

In addition, NONO has been implicated in transcriptional activation and repression, mRNA transport, and cortical neuronal migration [92,109,110]. More recently, hemizygous loss-of-function variants in the NONO gene have been associated with X-linked intellectual disability and global developmental delay. This X-linked recessive disorder is typically characterized by macrocephaly, elongated face, widely spaced eyes, and short forehead [78,79,80]. A novel *NONO* gene variant (Pro459Ala) has been identified in three male patients, each presenting a range of clinical symptoms. The broad phenotypic spectrum associated with this variant suggests a mild loss-of-function effect, potentially contributing to variable expression [111]. Based on the wide range of regulatory roles of NONO, the phenotypes observed in human patients, as well as in NONO-deficient mice, are likely the cumulative result of disrupted NONO-mediated nuclear paraspeckle assembly.

**SFPQ** (**s**plicing **f**actor **p**roline/glutamine(**Q**)-rich) is present in the cytoplasm of distal axonal regions and growth cones. It is also required for the regulation of gene transcripts during axonal growth in spinal cord motor neurons (Figure 1, Table 1) [112]. Recent findings have refined our understanding of how its low-complexity prion-like domains influence phase separation, revealing that the two low-complexity regions (LCR) exert distinct—and potentially opposing—effects on condensate formation. Notably, the shorter C-terminal LCR is the primary driver of condensation, while the longer N-terminal LCR, which constitutes roughly one-third of the protein and includes a prion-like domain, suppresses condensate formation in the full-length protein. These observations support a model showing that interactions between the two LCRs modulate LLPS [81,85]. Interestingly, loss of SFPQ gene function leads to embryonic lethality, likely due to its essential role in neuronal development. Conditional SFPQ-deletion mutant mice exhibit abnormal cerebral cortex development, indicating that SFPQ is critical for proper brain organization [84]. In addition, several genetic studies in ASD have identified SFPQ as a putative causative gene [113].

Taken together, nuclear paraspeckles are stress-induced structures assembled through liquid–liquid phase separation, involving NEAT1/2 long non-coding RNAs and RNA-binding proteins such as NONO and SFPQ. Mutations in these nuclear paraspeckle components have been associated with neurodevelopmental conditions, including ASD and intellectual disability.

## 4. RNA-Based Processing Bodies (P-Bodies/GW-Bodies)

Processing bodies (P-bodies) are present in some physiological conditions in cells but are also dynamically assembled in response to cellular stress that leads to the inhibition of translation initiation. P-bodies are constitutively associated with proteins involved in translational repression and mRNA decay (Figure 1). These include components of the cytoplasmic deadenylase complex, de-capping coactivator and enzyme, de-capping activators, and the 5′–3′ exoribonuclease [114,115,116]. P-bodies have also been shown to exhibit liquid droplet-like properties. In yeast cells, P-bodies are typically visible only upon stress induction and display liquid droplet phenotypes [41,117].

**TNRC6B** (**t**ri**n**ucleotide **r**epeat-**c**ontaining gene **6B** protein) is an RNA-binding protein and a key component of the miRNA-induced silencing complex (miRISC) involved in post-transcriptional gene regulation (Figure 1, Table 1). Upon LLPS, TNRC6B facilitates compartmentalization and regulation of gene silencing by organizing mRNAs and silencing factors into membraneless functional domains [87]. LLPS is driven by multivalent interactions between the GW-rich domain of TNRC6B and the tryptophan-binding pockets within the PIWI domain of Argonaute2 (Ago2). These condensates actively recruit de-adenylation factors and isolate target RNAs from the surrounding cytoplasmic environment, thereby enhancing the efficiency of miRNA-mediated silencing. These findings support a functional role for protein-mediated phase separation in facilitating miRNA-guided post-transcriptional repression [87].

Granadillo and colleagues reported a cohort of 17 patients with predicted damaging variants in TNRC6B, all of whom exhibited developmental delays beginning in early to mid-childhood, especially affecting speech. Approximately two-thirds of the patients also exhibited hypotonia. Over time, most individuals developed behavioral symptoms, including autism spectrum traits, ADHD, impulsiveness, and aggressive behavior, with many requiring therapeutic intervention and medication [86], further supporting the role of P-bodies in NDDs.

## 5. Conclusions

Although neurodevelopmental disorders manifest as dysfunction in neural networks (such as excitation/inhibition imbalance, dendritic arborization, and axonal growth), they often originate from more fundamental biochemical mechanisms. Recent studies underscore the central role of liquid–liquid phase separation in organizing membraneless organelles, such as stress granules, nuclear paraspeckles, and P-bodies. Understanding the molecular interactions and regulatory mechanisms of stress-induced membraneless organelles in neurons offers critical insight into the pathogenesis of these disorders and may inform future therapeutic strategies.

Our review has focused on collecting data on stress-induced membraneless organelle formation and how its disturbed functions can link with neurodevelopmental defects by highlighting potential links where the liquid–liquid phase separation-related properties of certain proteins overlap with their known involvement in neurodevelopmental disorders. On the other hand, liquid–liquid phase separation and neurodevelopmental disorders may both arise as independent functional outcomes of genetic mutations, and their co-occurrence does not necessarily imply a causal relationship [118,119]. However, in several instances—particularly where mutations impact intrinsically disordered regions that facilitate phase separation—emerging evidence suggests that liquid–liquid phase separation behavior may indeed be altered in patients [120]. Therefore, it is important to emphasize the need for future studies to reveal causative links between liquid–liquid phase separation and neurodevelopmental disorders, with the potential for developing new therapeutical strategies.

## Figures and Tables

**Figure 1 ijms-26-09068-f001:**
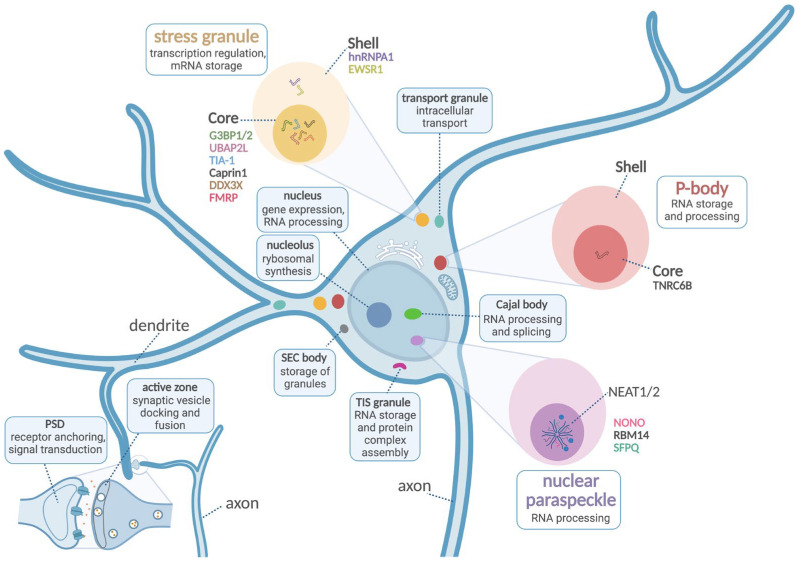
**Schematic illustration of phase-separated membraneless organelles in the neuron.** Boxes indicate the name and function of MLOs. The enlarged images (stress granule, P-body, and nuclear paraspeckle) present stress-induced membraneless organelles, with their phase-separated proteins indicated by different colors. Abbreviations: PSD—**p**ost**s**ynaptic **d**ensity, **s**uper **e**longation **c**omplex (SEC) body; TIS granule—**T**PA-**i**nducible **s**equence 11 protein family. Created in BioRender. Kimsanaliev, D. (2025) https://BioRender.com/jdt8ap6, accessed on 15 September 2025.

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
