# Peer review of "Stress-Induced Membraneless Organelles in Neurons: Bridging Liquid–Liquid Phase Separation and Neurodevelopmental Dysfunction"

_ijms, 2025, doi:10.3390/ijms26189068_

Round 1

Reviewer 1 Report

Comments and Suggestions for Authors

In this review article, Bencsik et al. summarized recent advances in liquid-liquid phase separation (LLPS) in neurodevelopmental disorders (NDDs), primarily focusing on stress-related membraneless organelles (MLOs): stress granules, nuclear paraspeckles, and P-bodies in neurons. There are numerous studies and reviews concentrating on how LLPS contributes to neurodegenerative diseases, but less focus on NDDs. It’s nice to see a good piece summarizing how LLPS play a role in NDDs. Overall, this review is well-organized and clearly expressed. I have only a few minor points for the authors to consider.

  1. The relationship between LLPS and NDD is vague. Generally, the authors emphasize the role of one gene from two aspects: the first is the ability to undergo LLPS; the second is genetic knockout (KO) or patient-associated variants that lead to NDD. But both LLPS and NDD can be functional consequences of a specific gene, and this does not suggest a causal relationship between LLPS and NDD. For example, whether a patient variation perturbs LLPS? Or does an LLPS mutation cause NDD phenotypes? If there is a lack of direct evidence, the authors should clearly point out current challenges and appeal to future studies.
  2. Regarding the “core” of the stress granule. Line 139, the authors concluded that “these assemblies are not purely formed via liquid-liquid phase separation”. But G3BP1, the major component of the “core”, is organized via LLPS. There is accumulating evidence showing a multi-phase organization pattern.
  3. Line 56, “The global prevalence of NDDs varies widely, ranging from 4.7% in Scotland [10] to 88.5% in Japan [11].” The 88.5% is astonishing, and I failed to find the source in reference 11.
  4. The majority of references are new, but some are inaccurate. For example, Line 77, “These structures enable localized regulation of processes in neurons, such as synaptic vesicles clustering [22,23]” ref 22 does not study synaptic vesicles. L82, “neurodegenerative diseases (e.g., amyotrophic lateral sclerosis, Alzheimer's disease and Parkinson’ disease) [27]”. Ref 27 is not a suitable one highlighting LLPS in neurodegenerative diseases. L120, “stress-induced MLOs has been linked to persistent protein aggregation and to the onset of NDDs in neurons [39].” Ref 39 is focusing on neurodegeneration rather than neurodevelopment.

Author Response

Response to Reviewer 1

We are grateful for your positive evaluation of our manuscript and for the correct and helpful comments. We were flattered to read that you rated our work well-organized and address your comments below. We hope that our response will be satisfactory and the modiefied manuscript will be suitable for publication.

  • The relationship between LLPS and NDD is vague. Generally, the authors emphasize the role of one gene from two aspects: the first is the ability to undergo LLPS; the second is genetic knockout (KO) or patient-associated variants that lead to NDD. But both LLPS and NDD can be functional consequences of a specific gene, and this does not suggest a causal relationship between LLPS and NDD. For example, whether a patient variation perturbs LLPS? Or does an LLPS mutation cause NDD phenotypes? If there is a lack of direct evidence, the authors should clearly point out current challenges and appeal to future studies.

Thank you for raising this important point. We fully agree that current evidences do not yet establish a direct causal relationship between LLPS and NDD. Our aim was to highlight potential links where LLPS-related properties of certain proteins overlap with their known involvement in NDDs. As you mentioned, LLPS and NDD may both arise as independent functional outcomes of genetic mutations, and their co-occurrence does not necessarily imply a causal relationship (Harrison and Shorter; 2017; Sprunger and Jackrel, 2021). However, in several instances—particularly where mutations impact intrinsically disordered regions that facilitate phase separation—emerging evidence suggests that LLPS behaviour may indeed be altered in patients (Mensah et al., 2023).

In light of this, we have revised our manuscript to more precisely state the current challenges in establishing causality and emphasize the need for future studies that directly assess whether disorders-associated mutations perturb LLPS. We hope that the modified text (line 408-420) adequately address your concerns.

  • Regarding the “core” of the stress granule. Line 139, the authors concluded that “these assemblies are not purely formed via liquid-liquid phase separation”. But G3BP1, the major component of the “core”, is organized via LLPS. There is accumulating evidence showing a multi-phase organization pattern.

Thank you for your feedback regarding the LLPS of G3BP1 protein. We have refined the liquid–liquid phase separation behaviour of G3BP1. In the revised text (line 154-157), we included the followings:

“ G3BP1 plays a pivotal role in organizing core and shell regions: its interactions with RNA and other SG components mediate the shell's rapid exchange dynamics, while multivalent interactions—particularly via IDRs and the RG-rich domain—contribute to the formation of dense cores [43,44]. ”

  • Line 56, “The global prevalence of NDDs varies widely, ranging from 4.7% in Scotland [10] to 88.5% in Japan [11].” The 88.5% is astonishing, and I failed to find the source in reference 11.

Thank you for pointing out this technical mistake, for which we sincerely apologize. We have corrected the the revised manuscript to ensure accuracy, and it reads now as (line56-59): “. The global prevalence of NDDs varies widely, due to the largely varying methodol-ogy of the studies, such as the different evaluation and characterization of affected children, as well as the level of activity of the countries in detection and diagnosis of NDDs [9]. Neurodevelopmental disorders showed the highest estimated prevalence among children younger than 15 years, varying from 4.1% in those under 5 years old to 7.0% in the 10 to 14 age group [10].”

  • The majority of references are new, but some are inaccurate. For example, Line 77, “These structures enable localized regulation of processes in neurons, such as synaptic vesicles clustering [22,23]” ref 22 does not study synaptic vesicles. L82, “neurodegenerative diseases (e.g., amyotrophic lateral sclerosis, Alzheimer's disease and Parkinson’ disease) [27]”. Ref 27 is not a suitable one highlighting LLPS in neurodegenerative diseases. L120, “stress-induced MLOs has been linked to persistent protein aggregation and to the onset of NDDs in neurons [39].” Ref 39 is focusing on neurodegeneration rather than neurodevelopment.

Thank you for noticing these inaccurete references in the text. We have corrected them according to your suggestions.

References

  • Harrison, A.F.; Shorter, J. RNA-binding proteins with prion-like domains in health and disease. Biochem J 2017, 474, 1417–1438, doi:10.1042/BCJ20160499.
  • Sprunger, M.L.; Jackrel, M.E. Prion-Like Proteins in Phase Separation and Their Link to Disease. Biomolecules 2021, 11, doi:10.3390/biom11071014.
  • Mensah, M.A.; Niskanen, H.; Magalhaes, A.P.; Basu, S.; Kircher, M.; Sczakiel, H.L.; Reiter, A.M.V.; Elsner, J.; Meinecke, P.; Biskup, S.; et al. Aberrant phase separation and nucleolar dysfunction in rare genetic diseases. Nature 2023, 614, 564–571, doi:10.1038/s41586-022-05682-1.

Reviewer 2 Report

Comments and Suggestions for Authors

This was a well-organized and easy-to-read review, and I learned a great deal from it. I have one point I'd like to confirm, which is shown below.

The prevalence rate for Japan in line56 seems too high to be correct, even by common sense. I briefly checked the cited references but found no such notation. Since this is a sensitive topic, if you're certain it's not wrong, provide more citations. One more point I'd like to raise: since the citations represent extractive study and cannot be considered comprehensive, it may not be accurate to present them as gaps between countries. Furthermore, the analysis is based on impressions from teachers and parents. Is this a credible paper that demonstrates strict prevalence rates? Either more citations should be added, or the phrasing should be made more restrictive.

Author Response

Response to Reviewer2

We are grateful for your positive review of our manuscript and hope that the revised version of our manuscript will enhance the overall value of the publication.

  • The prevalence rate for Japan in line56 seems too high to be correct, even by common sense. I briefly checked the cited references but found no such notation. Since this is a sensitive topic, if you're certain it's not wrong, provide more citations. One more point I'd like to raise: since the citations represent extractive study and cannot be considered comprehensive, it may not be accurate to present them as gaps between countries. Furthermore, the analysis is based on impressions from teachers and parents. Is this a credible paper that demonstrates strict prevalence rates? Either more citations should be added, or the phrasing should be made more restrictive.

Thank you for pointing out this technical mistake, for which we sincerely apologize. We have corrected the the revised manuscript to ensure accuracy, and it reads now as (line56-59):” . The global prevalence of NDDs varies widely, due to the largely varying methodol-ogy of the studies, such as the different evaluation and characterization of affected children, as well as the level of activity of the countries in detection and diagnosis of NDDs [9]. Neurodevelopmental disorders showed the highest estimated prevalence among children younger than 15 years, varying from 4.1% in those under 5 years old to 7.0% in the 10 to 14 age group [10].”